# "Merge Conflicts!" Exploring the Impacts of External Knowledge Distractors to Parametric Knowledge Graphs

**Cheng Qian**[1,2*], **Xinran Zhao**[3], **Sherry Tongshuang Wu**[3]
[1]Tsinghua University, [2]University of Illinois Urbana-Champaign,
[3]Carnegie Mellon University
chengq9@illinois.edu

## Abstract

Large language models (LLMs) acquire extensive knowledge during pre-training, known as their *parametric knowledge*. However, to remain up-to-date and align with human instructions, LLMs inevitably require *external knowledge* during interactions. This raises a crucial question: How will LLMs respond when external knowledge interferes with their parametric knowledge? To uncover the impacts systematically, we construct *parametric knowledge graphs* to reveal different LLM knowledge structures, and introduce external information through *external knowledge distractors* of varying degrees, methods, positions, and formats. Experiments on both closed and open-source models demonstrate that LLMs tend to believe in external knowledge sources, particularly when they direct conflict or make confounding changes within detailed contexts. We also discover while LLMs are sensitive to external knowledge veracity, they still get distracted by unrelated information. These findings highlight the mechanisms behind LLM's integration of external knowledge, even indirectly, during model-user interactions.

## 1 Introduction

Current large language models (LLMs) have assimilated extensive knowledge during pre-training (Chowdhery et al., 2022; Thoppilan et al., 2022; OpenAI, 2022; 2023; Touvron et al., 2023; Anil et al., 2023; Zeng et al., 2022), converting it into *parametric knowledge*. However, LLMs still struggle to stay current with world events and often require background information in real-world applications (Trivedi et al., 2023; Yu & Ji, 2023). This necessitates the use of *external knowledge*, which can be incorporated either explicitly through retrieval-augmented generation (RAG) methods (Lewis et al., 2020; Chen et al., 2017) that retrieve knowledge from sources like databases or documents (Shi et al., 2023; Ram et al., 2023), or via tools that provide access to APIs and online resources (Schick et al., 2023; Qin et al., 2023); or implicitly through carefully designed prompts and human-provided instructions.

However, the introduction of external knowledge may present a direct conflict with LLM's parametric knowledge (Xie et al., 2023; Neeman et al., 2022), caused by information updates, related misinformation, or fictional information, respectively illustrated on the left of Figure 1. Regardless of categories, these conflicts nevertheless will cause instability in LLM's beliefs and responses, and the interconnected nature of LLM's parametric knowledge (Petroni et al., 2019; Wang et al., 2020) may cause further indirect interfering effects. For the instance in Figure 1, as we introduce updated knowledge only to the $2^{nd}$ hop through prompting ("The PM of UK is Rishi Sunak"), the model's response to the $3^{rd}$ hop (final) also shifts from the *Year 2019* to *Year 2022*.

This phenomenon has recently been called the "ripple effect" (Cohen et al., 2023). Previous works have proposed benchmarks (Zhong et al., 2023) and metrics (Cohen et al., 2023) for its evaluation, yet i) they typically focus on linear relations among closely connected knowledge

---

\* Work done as a student intern at CMU.

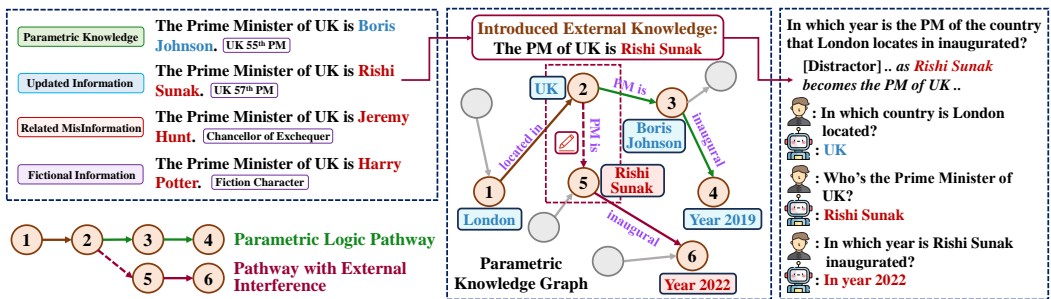

Figure 1: Information updates on UK's PM bridges a new relation that diverges final answer.

entities and ii) manual efforts are required to construct specific external knowledge and assess the extent of the ripple effect.

To tackle these issues, we present a framework for *systematically assessing the interactions between parametric and external knowledge.* Inspired by well-structured knowledge graphs (KGs), we introduce the *parametric knowledge graph* (PKG), which automatically extracts the LLM's interconnected parametric knowledge into flexible graphs with hundreds of entities and relations. In Figure 1, nodes and solid lines illustrate a PKG sub-graph, featuring entities like countries, political figures, and relations such as "located in". Building upon PKG, we further define *distractors* – external knowledge introduced through prompts with varying degrees, methods, positions, and formats, thus encompassing different conflict types while all interfering with the existing knowledge in PKGs. Our definition allows for direct investigation into interactions between external information and PKGs: in Figure 1, the distractor representing updated information (UK, PM is, Rishi Sunak) bridges a new relation between PKG nodes, initiating a ripple effect that deviates the model response.

We investigate this interaction via experiments involving both black-box GPT3.5 and open-source MPT-7B. By first introducing the distractor to the model, we perform queries interactively in a one-hop manner, on the right of Figure 1. We assess the model's responses in terms of *consistency* (adherence to its PKG) and *confidence* (the probability of providing this answer) when distractors are present. In our observations, LLMs tend to deviate from their parametric knowledge when they lack confidence in it initially. Interestingly, they also consistently exhibit higher confidence in responses when confronted with external knowledge. Analyzing the effects of various distractor types, we discover presenting direct conflicts or confounding changes instead of evidently false information wields greater influence. We are also surprised by many findings, e.g. *weak* distractors that don't directly alter the model's logic pathway can impact responses; The impact of distractors can be enhanced if embedded in lengthier contexts; GPT-3.5 and MPT-7B display distinct patterns in their susceptibility to different distractors. All these findings underscore the mechanisms behind LLM's integration of external conflicting knowledge, even indirectly, during its active model-user interactions.

## 2  Related Work

**Causes and Solutions to Knowledge Conflicts.** LLMs amass extensive parametric knowledge through pre-training (Roberts et al., 2020; Jiang et al., 2020; Gururangan et al., 2020), weaving a unique system of knowledge. However, inaccurate or outdated data may lead to hallucinations (Carlini et al., 2021; Lazaridou et al., 2021; Zhang et al., 2021), thereby driving the employment of tools (Schick et al., 2023; Qin et al., 2023), memory techniques (Zhong et al., 2022), and retrieval strategies (Guu et al., 2020; Izacard & Grave, 2021) to align the model. However, such external knowledge may interfere with the existing parametric knowledge, caused by information updates (Zhang et al., 2024; Yu & Ji, 2023), noise in retrieved knowledge (Cuconasu et al., 2024; Chen et al., 2023), etc. Existing strategies include knowledge editing for updates (Mitchell et al., 2021; De Cao et al., 2021; Hase et al., 2021), disentanglement of responses corresponding to knowledge sources (Neeman et al., 2022), and utilization of abstention to improve faithfulness (Zhou et al., 2023). In line with

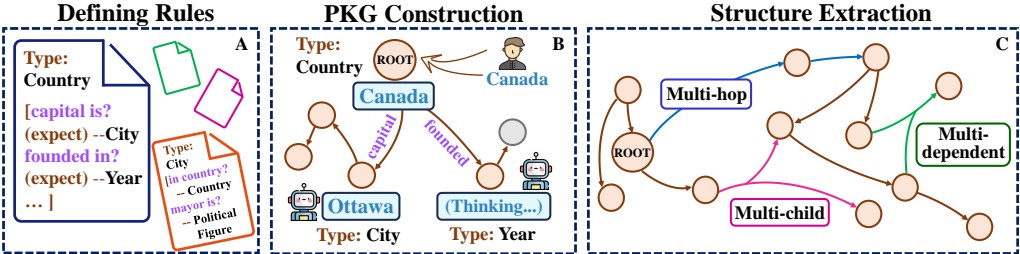

Figure 2: **A, B**: Construct PKG with defined rules automatically. **C**: Extract various PKG structures.

these studies, we provide a unified framework to study the conflicts, encompassing various conflict types and shedding light on strategies.

**Propagation of Introduced Knowledge.** Prior approaches to integrate external knowledge centered on modifying parameters (Meng et al., 2022; Yao et al., 2022) and incorporating specialized modules (Wang et al., 2021; Mitchell et al., 2021). Nevertheless, introduced external knowledge may exert long-lasting effects. Onoe et al. (2023) discovered that prepending entity definitions can facilitate the propagation of the injected external knowledge. Zhong et al. (2023) contributed a benchmark to measure how the alteration of knowledge may influence the entire multi-hop queries. This phenomenon is termed the ripple effect by Cohen et al. (2023), who offer six metrics to assess the robustness of the model editing methods. Building on these studies, our framework systematically extends focus on multiple parametric knowledge structures and external impacts.

# 3 Introduction of Parametric and External Knowledge

Knowledge conflicts arise in various ways including knowledge updates, noisy context retrieval, etc. However, existing studies lack a unified platform to study the types and effects of external knowledge. To fully uncover its impacts, we construct *parametric knowledge graph* (PKG) to capture the relations of the model's internal belief (Section 3.1), and introduce *external knowledge distractors* (EKDs) of different categories for systematic evaluation (Section 3.2). These introduced knowledge will cover multiple dimensions, including updated information, misinformation, and fictional information, to imitate different knowledge conflict scenarios that may occur in real-world applications and enhance the generalizability of our framework.

## 3.1 Parametric Knowledge Graph

To exploit the model's interconnected parametric knowledge, we first propose to construct its *parametric knowledge graph* (PKG). Similar to a knowledge graph (KG), a PKG consists of nodes ($E$) representing entities and edges ($R$) representing relations, converting the model's *implicit* knowledge into *structured* representations. Unlike traditional KGs, PKG is grounded in the model's knowledge rather than real-world facts (Fensel et al., 2020).

**Construction.** PKG construction is automated using *specified rules*. Each entity $E$ (e.g., France) in PKG is abstracted into a *type* ("Country"), with rules created for each type as ($R$, *target type*). For instance, the type "Country" has the relation "capital is", targeting type "City" as the answer. Rules are implemented in natural language templates (Appendix A), mapping LLM's logic pathways to the graph interpretably. Specifically, given a root node (an entity with its type), the PKG is extended depth-first. As in Figure 2B, assigning "Canada" as the root node leads to extensions of all relations that type "Country". has. The model then seeks answers for each target type, recursively forming the PKG. Only answers consistent across consecutive queries are considered as parametric knowledge (Appendix A).

**Extraction.** A PKG's key advantage includes *extraction of data chains with structural variety* (Figure 2C), including multi-hop, multi-child (multiple answers for a given entity and relation), and multi-dependent (two entities jointly decide the answer for a relation). In

| A. Multi-Hop | | | | B. Multi-Dependent | | | |
|---|---|---|---|---|---|---|---|
| Structure | Name | Hops | Example | Structure | Name | Hops | Example |
| | 2-Hop Structure | 2 | Who's PM of the country that **London** locates in? | | 1-1-0 Structure | 3 | Who is the PM of the country to which **London** belongs in the year when **Michael Jackson** is born? |
| | 3-Hop Structure | 3 | In which year is PM of the country that **London** locates in inaugurated? | | 1-1-1 Structure | 4 | What's the birthplace of the PM of the country to which **London** belongs in the year when **Michael Jackson** is born? |
| | 4-Hop Structure | 4 | Who's the US president in the year that PM of the country that **London** locates in inaugurated? | | 1-2-0 Structure | 4 | Who is the PM of the country to which **London** belongs in the year when the singer of album **Dangerous** is born? |

Table 1: Multi-hop and multi-dependent structures investigated in experiments. The red and blue nodes denote starting entities. The green edges denote the multi-dependent relations contained in the *pivot hop*, while purple edges denote other explicit relations.

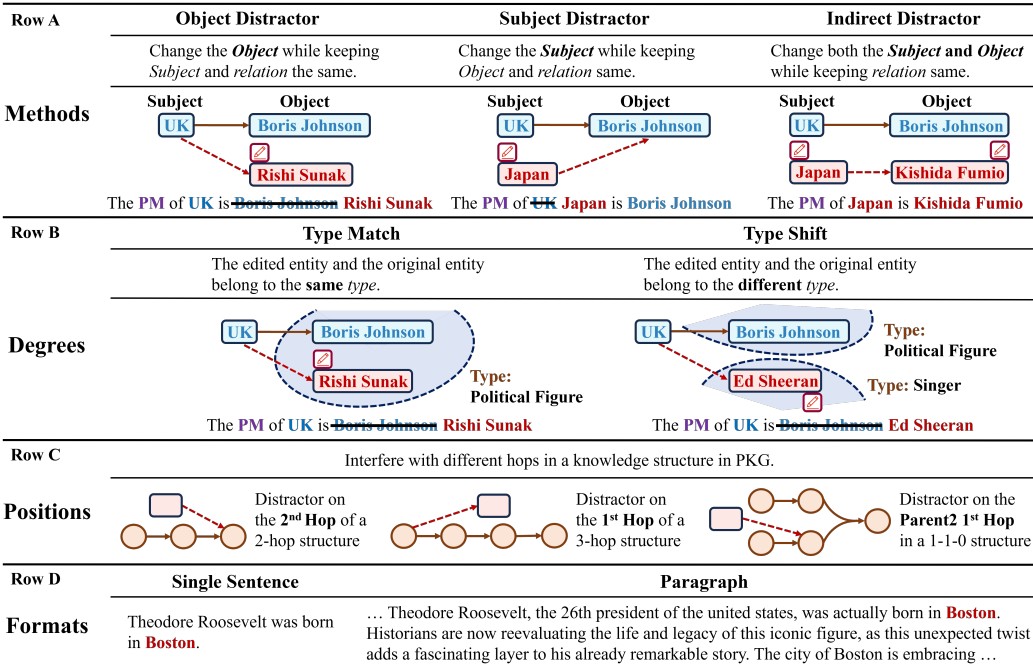

Figure 3: Illustration of different distractors types applied in experiments.

experiments, we extract PKG sub-graphs with different structures, linearizing each into a "data chain" represented by triplets $[(E_0, R_1, E_1), (E_1', R_2, E_2), ..., (E_{n-1}', R_n, E_n)]$ ($E_k = E_k'$ for multi-hop chains). We primarily use multi-hop structures as query bases (Table 1A). To encapsulate non-linearity, we further include three multi-dependent structures (Trivedi et al., 2022), each containing a multi-dependent relation denoted as the *pivot hop* (Table 1B). The pivot hop depends on two upstream entities, both of which can end multi-hop chains of lengths $A$ and $B$. Simultaneously, the pivot hop's answer entity can start a multi-hop chain of length $C$. We use $A$-$B$-$C$ to denote the multi-dependent configuration.

## 3.2 External Knowledge Distractors

To comprehensively evaluate knowledge conflicts, we further introduce *external knowledge distractors* (EKDs) obtained by modifying raw data chains extracted from PKGs. We employ GPT-3.5 for their automatic construction (Appendix B), and EKDs are provided in natural language descriptions before model-user interactions (Figure 1).

We systematically construct EKDs by manipulating four dimensions (Figure 3): methods, degrees, positions, and formats. The edited external information may contain updated information or non-factual details, but the primary purpose is to interfere with the model's

original beliefs, which is the crucial attribute of EKDs we focus on. Through careful constructions, we aim to cover different types of knowledge conflicts and reflect realistic scenarios, while enabling in-depth analysis of how the model handles and responds to these conflicting knowledge inputs.

**Methods.** EKD methods illustrate the relationship between external and parametric knowledge. As in Figure 3A, the *Object Distractor* alters the object in the raw data chain. For instance, replacing "Boris Johnson" with "Rishi Sunak" results in an updated knowledge "The PM of UK is Rishi Sunak," an explicit contradiction to the model's original belief. Similarly, the *Subject Distractor* and *Indirect Distractor* modifies the subject or both the subject and object, respectively. Relations are kept the same for all three methods.

**Degrees.** EKD degrees reflect the severity of deviation from the parametric knowledge (Figure 3B). The deviation is categorized as either *Type Match* or *Type Shift* depending on whether the edited and original entities belong to the same type. *Type Match* EKDs (e.g., political figure "Boris Johnson" to "Rishi Sunak") are usually more confusing, while *Type Shift* EKDs (e.g., political figure "Boris Johnson" to singer "Ed Sheeran") are generally more apparent and lack credibility.

**Positions.** EKD positions indicate where this external information is introduced in the data chain (Figure 3C), rather than how the knowledge is modified. The number of places to introduce EKDs is decided by the number of hops in the extracted PKG structure. Different positions represent different stages during the model-user interaction.

**Formats.** EKD formats are differentiated by context length, thereby forming *Single Sentence* EKDs and *Paragraph* EKDs. Figure 3D illustrates how a simple piece of knowledge can be extended into a paragraph. The purpose of this is to test if the model exhibits bias towards more detailed descriptions.

## 4  Experiment Setup

To evaluate the model's reactions when external knowledge interferes with its parametric knowledge, we conduct systematic experiments by varying the combinations of model PKG structures and EKD types. Appendix C provides more details on hyper-parameters and settings.

**Method.** To inspect the model's responses during active interactions instead of testing its multi-hop reasoning ability, we follow the "instance-wise" probing method proposed by Zhong et al. (2023). As shown in Figure 1, we first present the EKD to the model as external knowledge, and then probe the model's answers in a one-hop manner. Following the data chain $[(E_0, R_1, E_1), (E'_1, R_2, E_2), ..., (E'_{n-1}, R_n, E_n)]$ extracted from PKG, we probe for the model's answer $A_1$ after inquiring $(E_0, R_1)$. Then we continue to probe the model's answer $A_2$ after inquiring $(A_2, R_1)$. This iterates until all queries are done or the model abstains from answering.

| **Rules** | Total Types | 17 | |
| | Total Rels | 63 | |
| **Dimensions / Model** | | **GPT3.5** | **MPT-7B** |
| Avg Node Num | | 278 | 166 |
| Avg Edge Num | | 467 | 276 |
| Multi-dependent Rels | | 769 | 443 |
| Multi-child Rels | | 192 | 124 |
| Multi-hop Structures | 2-hop | 5,361 | 3,360 |
| | 3-hop | 14,523 | 8,642 |
| | 4-hop | 28,297 | 17,064 |

Table 2: The statistics of 8 PKGs we apply respectively for GPT3.5 and MPT-7B. *Rels* denotes relations. The magnitude of relations and varied structures exemplify PKGs' heightened diversity and complexity.

**Models.** We experiment on the open-source MPT-7B (ML, 2023) and the black-box GPT3.5 (OpenAI, 2022) due to their robust interaction capabilities and the convenience of confidence analysis. Appendix E presents results from GPT3 as additional support to our findings.

**Data.** We construct 8 different PKGs for both GPT3.5 and MPT-7B, with statistical findings detailed in Table 2. For all studies besides PKG knowledge structures, we apply 200 chains

| EKD Types | | GPT3.5 Consistency (%) | MPT-7B Consistency (%) | Confidence (%) | Conclusions |
|---|---|---|---|---|---|
| **A:** | | **Multi-hop Structures (2 / 3 / 4 hops), Macro Avg.** | | | |
| **Degrees** | Type Match | 55.90 | 42.78 | 80.92 | Models resist EKDs that evidently lacks veracity, but they still make the model more uncertain generally. |
| | Type Shift | 58.50 | 46.22 | 78.35 | |
| **Methods** | Object | 42.90 | 35.71 | 80.53 | *Object Distractors* most easily mislead the models, but even indirect "weaker" EKDs exhibit interference. |
| | Indirect | 60.08 | 43.97 | 77.20 | |
| | Subject | 67.62 | 54.17 | 81.22 | |
| **Positions** | 1st Hop | 61.78 | 42.11 | 78.68 | GPT3.5 exhibits defense against EKDs in the first hop, while MPT-7B easily believes in EKDs in the beginning. Consistency generally rises as the interaction evolves (explicit in later analysis). |
| | 2nd Hop | 53.53 | 45.25 | 80.77 | |
| | 3rd Hop | 54.00 | 43.25 | 79.17 | |
| | 4th Hop | 54.00 | 40.17 | 78.67 | |
| **Formats** | Sentence | 57.20 | 44.51 | 79.65 | Lengthier and more detailed contexts in EKDs lower the model's consistency. |
| | Paragraph | 54.37 | 42.06 | 78.26 | |
| **B:** | | **Multi-dependent Structures (1-1-0, 1-1-1, 1-2-0), Macro Avg.** | | | |
| **Degrees** | Type Match | 48.55 | 32.89 | 78.60 | The conclusions for multi-dependent structures on EKD degrees and methods remain the same as those for multi-hop structures. |
| | Type Shift | 50.23 | 35.12 | 76.81 | |
| **Methods** | Object | 36.40 | 24.72 | 77.70 | |
| | Indirect | 49.58 | 33.29 | 76.51 | |
| | Subject | 62.19 | 44.00 | 78.91 | |
| **Positions** | Parent1 1st Hop | 54.83 | 30.56 | 77.07 | The *pivot hop* exhibits special traits. Introducing EKDs to interfere with the *pivot hop* results in the lowest consistency for GPT3.5 while highest consistency for MPT-7B. |
| | Parent2 1st Hop | 54.22 | 31.72 | 75.84 | |
| | Parent2 2nd Hop | 51.00 | 32.50 | 78.52 | |
| | Pivot Hop | 39.44 | 39.22 | 80.20 | |
| | Child 1st Hop | 49.33 | 37.83 | 78.67 | |

Table 3: Experimental results and conclusions on the impacts of various EKDs to multi-hop and multi-dependent PKG structures. The results we focus on are shaded: green for the *highest* numerical value of an EKD type, while red for the *lowest*. Appendix D provides more detailed results.

for each type of N-hop data chain ($N \in 2, 3, 4$). For the study on PKG knowledge structures, we extract 100 raw data chains for each multi-dependent structure, illustrated in Table 1B.

**Metrics.** We use **consistency** to measure whether the model always sticks to the answer in PKG during multiple rounds of queries. Formally, among $N$ query chains $C_1, ..., C_N$, the model outputs the final answer that adheres to PKG in $M$ chains. *Consistency* is defined as:

$$\text{Consistency}(\{C_1, ..., C_N\}) = \frac{M}{N}. \tag{1}$$

We also compute **confidence** to quantify MPT-7B's likelihood of outputting the target entity. Formally, given the tokens $t_1, ..., t_M$ of the entity $E$ in the model's response, the model's *confidence* in outputting this entity as the answer is defined as:

$$\text{Confidence}(E) = \prod_{t=t_1}^{t_M} \frac{e^{z_t}}{\sum_{i=1}^{N} e^{z_i}}, \tag{2}$$

where $z_t$ denotes the raw score (logit) associated with the token $t$, and $N$ denotes the total number of tokens in the vocabulary.

In addition, we investigate whether the model's response for a specific hop of query aligns with its PKG by classifying it into either *conforming* (when it aligns with PKG) or *deviated* (when it is derived from the EKD).

# 5 Experiment Result

## 5.1 Effectiveness of EKDs through Confidence Analysis

We first analyze in general *why the EKDs we introduce are effective*. Through the lens of confidence, we aim to unveil the mechanism behind the model responses under interference.

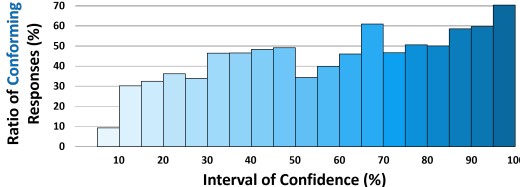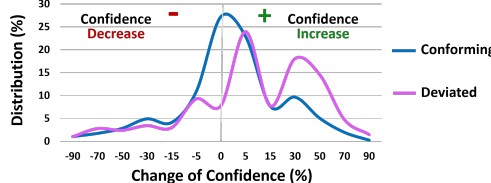

Figure 4: **Left**: The ratio of conforming responses concerning the confidence placed in the corresponding relations in PKG. **Right**: The distribution of the *change* of confidence after introducing the distractors respectively concerning conforming and deviated responses. The area under the curve left of 0 represents negative confidence change, and vice versa.

| EKDs (Method) | Object | Indirect | Subject |
|---|---|---|---|
| **Confidence (%)** | 77.70 | 74.31 | 71.90 |

Table 4: The average confidence of relations in PKGs that the model deviates in later responses. "*Weaker*" EKD methods tend to convince the model of knowledge that it is not confident about.

**Consistency occurs with high confidence.** With interference, the model's responses are more likely to conform with PKG when it is confident about this piece of PKG knowledge. As illustrated in the left of Figure 4, the low conforming response rate on the left also suggests that if the model's initial confidence in PKG knowledge is low, EKDs are more likely to cause deviations in subsequent queries.

**Response deviates with raised confidence.** The model's confidence generally increases with the presence of external knowledge, particularly for deviated responses. Through analyzing the change of confidence in the right of Figure 4, we reveal that: i) The area under positive confidence change is larger, indicating external knowledge generally boosts the model's confidence. ii) Most deviated responses have increased confidence, proving that EKDs can cause model deviation with higher confidence.

## 5.2 Results on Different EKD Types

Presented in Table 3A, we further investigate the impacts of EKD degrees, methods, positions, and knowledge formats. Appendix D presents more detailed results on different PKG structures, Student's t-test, and inconsistency analysis.

**Degrees: Models exhibit resistance to knowledge that evidently lacks veracity.** We discover that compared to *Type Match* EKDs, *Type Shift* EKDs are less successful in diverting the model's responses. In Table 3A, we show the consistency is always higher for *Type Shift* EKDs (P-value $p < 0.001$ in Student's t-test). As *Type Shift* EKDs change edited entity's type and often yield external knowledge beyond commonsense, our results demonstrate LLMs are resistant to such knowledge that lacks veracity.

Nevertheless, the confidence of model responses decreases for *Type Shift* EKDs, suggesting that while the model may reject them, their presence still exerts strong effects of uncertainty.

**Methods:** *Object Distractors* **yield the lowest consistency, while "weaker" EKDs also cause interference.** *Object Distractors*, which introduces direct conflicting external knowledge, particularly divert the model response from its PKG. As shown in Table 3A, they result in the lowest consistency (P-value $p < 0.001$). Among all methods, only *Object Distractors* maintain the original subject, thus they create new relation links that more easily diverge the

---

(GPT3.5) *Indirect Distractor:*
~~China~~ **US** established diplomatic relations with Turkey in ~~1971~~ **1947**.
*[… Interaction History …]*
**User:** What's the year when **China** the first time establish diplomatic relations with Turkey?
**GPT3.5: 1947**

(MPT-7B) *Indirect Distractor:*
~~Capitol Records~~ **Coca-Cola** CEO in 1948 is ~~Johnny Mercer~~ **Santa Claus**.
*[… Interaction History …]*
**User:** Who is the CEO of **Capitol Records** in the year 1948?
**MPT-7B: Santa Claus**

---

Figure 5: Case study on how the model deviates in response under the *Indirect Distractor*. The weak belief of their parametric knowledge in PKGs and some intrinsic similarity in details (e.g. 1948) mislead both models.

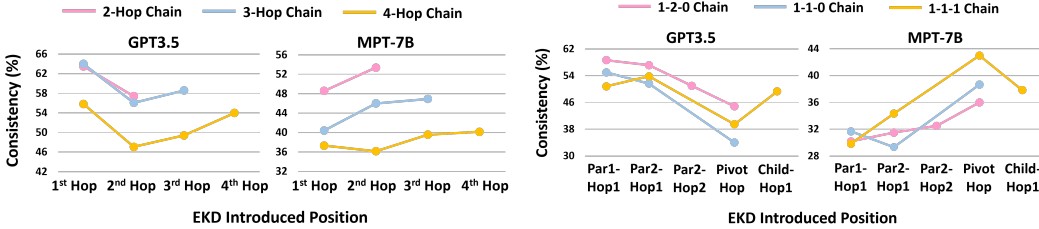

Figure 6: **Left**: The consistency concerning which position the EKD is introduced for 2, 3, and 4-hop chains. **Right**: The consistency concerning which position the EKD is introduced for multi-dependent structures. *Par* denotes the chains extended from the two parent nodes of the pivot hop.

model's logic pathways, making them especially convincing.

We note that *Subject* and *Indirect Distractors* also cause interference though they do not explicitly align with the query (thus are "weaker" EKDs). As shown in Table 4, they are more likely to divert the model responses in knowledge with weak initial belief in PKGs. Specifically, we present a case study in Figure 5: For GPT3.5, the uncertainty about the user's query drives it to extract "1947" from the distractor as the final answer, despite the subject in distractor being "US" rather than "China"' as inquired. The same happens to MPT-7B, as the same additional information "in the year 1948" presented in both the query and the distractor drives the model to trust "Santa Claus" as the company's CEO, though it is evidently false.

**Positions: Models resist EKDs as interaction evolves, and GPT3.5 defends against early introduced EKDs.** We differentiate three structures on the left of Figure 6, and discover both models' consistency increases as EKDs are introduced to interfere with later hops in the data chain. GPT3.5 also particularly resists external knowledge that interferes with the initial hop. The rising trend can be attributed to both models' declining attention to EKDs as the interaction progresses. The high consistency of GPT3.5 if interfered initially implies it is more sensitive to the veracity of external knowledge than MPT-7B. Appendix D.3 provides an additional ablation study to further support this finding.

**Formats: Lengthier contexts decrease consistency.** Both GPT3.5 and MPT-7B tend to trust longer and seemingly more compelling external knowledge. As shown in Table 3A, consistency decreases for both models when applying *Paragraph* EKDs with more details (P-value p ¡ 0.001). To understand why the model belief changes, we further investigate the interactions between formats and other EKD attributes.

| Metrics | | Conform Res. | | Deviated Res. | |
|---|---|---|---|---|---|
| | | Match | Shift | Match | Shift |
| **Change of** | 2-hop | -1.55 | -2.34 | -2.68 | +2.42 |
| **Confidence** | 3-hop | -1.25 | -0.69 | -0.50 | +5.69 |
| **(%)** | 4-hop | -0.94 | -0.21 | +0.51 | +0.40 |

Table 5: The *change* of confidence concerning EKDs of different degrees when the format becomes lengthier.

i) Formats×Degrees: *Detailed contexts increase belief in more severely edited external knowledge.* Table 5 shows model confidence increases for deviated responses to *Type Shift* EKDs, but decreases for all conforming responses. This suggests longer contexts lower model confidence in extracting a target entity as the answer, but increase trust in more severely edited external knowledge. ii) Formats×Methods: *Detailed contexts increase belief in "weaker" EKDs.* Table 6 shows model confidence generally increases for deviated responses to *Indirect*

| Metrics | | Conforming Res. | | | Deviated Res. | | |
|---|---|---|---|---|---|---|---|
| | | Obj. | Indir. | Sbj. | Obj. | Indir. | Sbj. |
| **Change of** | 2-hop | -2.21 | -1.55 | -2.13 | -4.52 | +3.58 | +1.80 |
| **Confidence** | 3-hop | -1.24 | -0.67 | -0.91 | -1.93 | -0.28 | +9.26 |
| **(%)** | 4-hop | -0.66 | -1.03 | -0.56 | -1.80 | -0.16 | +5.03 |

Table 6: The *change* of confidence concerning EKDs applying different methods when the format becomes lengthier.

and *Subject Distractors*. We conclude that lengthier contexts effectively make the model trust knowledge it previously distrusts.

### 5.3 Results on More PKG Structures

Previous analyses mainly employ multi-hop structures, focusing on one-to-one relations. Now, we incorporate multi-dependent structures to comprehensively study and evaluate EKD's impacts on diverse PKG sub-graphs.

**Consistent results for EKD methods and degrees.** Our earlier findings on EKD methods and degrees are reinforced by multi-dependent structures, with more details illustrated in Table 3B and Appendix D.5.

**Unique traits of pivot hop.** The impact of different EKD positions differs from previous findings due to changes in the underlying knowledge structure. For multi-dependent structures, GPT3.5 has the lowest consistency when interfered with the *pivot* hop, while MPT-7B has the highest. Similarly, we differentiate three multi-dependent structures on the right of Figure 6. The trend it illustrates suggests that GPT3.5 is more affected by EKDs with extra information, while MPT-7B is less so.

## 6 Discussions

**Effects of Indirect Interference.** While we can expect direct conflicts to cause the model's inconsistency, we are surprised that indirect interference can also convince the model (Figure 5). We identify two underlying reasons for this: i) **Veracity**: *Indirect Distractor* could be a fact, and the model's "faith" in correctness causes it to doubt its original answer. ii) **Matched Details**: Similar details in the query and external information lead the model to believe in a strong correlation between them, thus deviating from the original parametric answer. The impact of indirect interference suggests future studies on removing information snippets causing unexpected effects will be valuable.

**Bias Towards Lengthier Context.** Both GPT and MPT models show low consistency when knowledge is presented in lengthier formats, indicating their bias towards *persuasiveness* similar to human decision-making. Besides, the model is more inclined to accept previously unaware or doubtful external knowledge than blatantly false information, especially when provided in a detailed context. This bias raises concerns about potential misuse and suggests the need for verification. Methods that compare model behaviors on long prompts vs. short but equally informative prompts (e.g., a high-quality summarization) might be a useful layer for rectifying LLM outputs.

**GPT vs. MPT: GPT's Initial Distrust in External Knowledge.** We discover both GPT3 and GPT3.5 maintain high consistency if the interference is introduced to the initial hop of the data chain (appendix E.3). This distrust and vigilance is not observed in MPT-7B, suggesting differences in the training process or protection mechanisms. However, GPT models are most likely to deviate in the $2^{nd}$ hop, indicating decreased attention over time.

## 7 Conclusion

This paper investigates the effects of external knowledge on parametric knowledge via systematic experiments. We devise a framework to construct the model's *parametric knowledge graph* and corresponding *external knowledge distractors* for interference. We examine the impact of external knowledge's degrees, methods, positions, and formats on multi-hop and multi-dependent parametric knowledge structures. Our results on GPT3.5 and MPT indicate that responses tend to deviate from the original PKG when external information poses direct conflicts (*Object Distractors*), gives confounding changes (*Type Match* EKDs), or provides detailed context (*Paragraph* EKDs). We also find that GPT models are sensitive to external information's veracity in the beginning (*Interfered at $1^{st}$ Hop*), and both models are vulnerable to unrelated external knowledge (*Indirect Distractors*). These findings elucidate

how LLMs manage potential conflicts and suggest the mechanisms behind the LLM's incorporation of external knowledge, even implicitly. We hope our framework can provide a unified platform and facilitate further investigations into the interaction between external and parametric knowledge.

## Acknowledgement

This work was funded by N000142312840. The work was also partially funded by Tsinghua University.

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

# Appendix

## A  Details on Revealing Model's PKG

We reveal the model's PKGs using natural language templates, as we show in Figure 13. During the construction of PKGs, we query the model three times with different temperatures ($T = 0.3, 0.5$, and $0.7$). The prompt for retrieving the answer is shown in Figure 14. Then, we judge the consistency of the model's responses through the checker we present in Figure 15. If we finally get "N/A", then we move on to search for other relations complying with the rules that the models are more confident about. Otherwise, we add the relation and the model's consistent answer into the PKG, regarding it as a piece of the model's parametric knowledge.

## B  Details on Constructing EKDs

After extracting multiple structures as the original data chain from the model's PKGs, we perform modifications to the data chain to introduce EKDs as external knowledge. This process is automated with the help of GPT3.5. Among the four types of EKDs, methods and degrees both directly modify the original information. Three methods and two degrees combine into a total of six types of EKDs. The prompts applied for constructing these six types of EKDs are introduced from Figure 16 to Figure 21.

Upon getting these six types of EKDs, the external information we get is in a format of *Single Sentence*. To turn them into external knowledge presented in multiple sentences, we apply the prompt in Figure 22 to construct EKDs in *Paragraph* format. EKDs introduced to interfere with different positions do not need additional construction.

## C  Details on Experimental Settings

**Models.**   We apply both GPT3.5 and MPT-7B models. For all the experiments, we set *top-p* to 1 and *temperature* to 0.3. The same setting across all experiments guarantees fairness when we are measuring the model's consistency and ensures that the model's confidence is comparable. We set the max sequence length to 512, and for both models, we do not add the frequency or presence penalty.

**Controlled Settings.**   To control the variables in our experiment, for all the studies except knowledge structures, we experiment on all the *multi-hop* structures as raw data chains. For all the studies except the external knowledge format, we apply *Single Sentence* as the EKD's knowledge format. Please refer to appendix C for a more detailed explanation of each experimental setting.

**Division of Results.**   As introduced in appendix B, combining EKD methods and degrees, we get six different types of EKDs for each hop of the query (each data chain may have multiple hops of the query). We experiment with all these EKDs. For the results regarding degrees, we divide the results based on the two different EKD degrees. For the results regarding methods, we divide our results based on the three different methods applied to the construction of EKD. For the results regarding positions, we divide the results based on which hop of query in the knowledge structure the EKD is introduced to. For the results on knowledge formats, we introduce a *Paragraph* version to all previous EKDs and repeat all the experiments for comparison.

**Data.**   We construct 8 PKGs with different root nodes for both GPT3.5 and MPT-7B using manually defined rules. In total, we used 17 types and 63 relations in the construction rules. We present in Figure 7 more details on the types of entities in PKGs we apply. For all the studies besides the knowledge structures in PKG, we employ N-hop data chains ($N \in 2, 3, 4$), utilizing 200 chains for each type. Each $N$-hop data chain affords $N$ positions for external knowledge introduction, three different methods, and two different degrees,

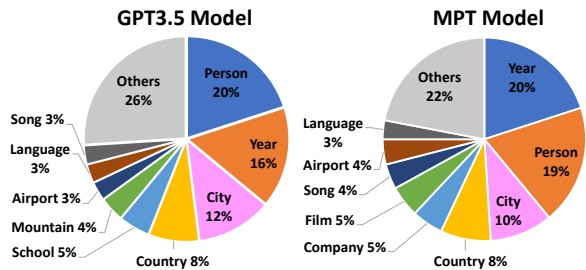

Figure 7: Ratio of different *types* in model's PKG.

| Metrics | | GPT3.5 | |
|---|---|---|---|
| | | Type Match | Type Shift |
| **Consistency (%)** | 2-hop | 59.92 | 61.00$_{\uparrow 1.1}$ |
| | 3-hop | 56.78 | 62.33$_{\uparrow 5.6}$ |
| | 4-hop | 51.00 | 52.17$_{\uparrow 1.2}$ |
| **Metrics** | | **MPT-7B** | |
| | | Type Match | Type Shift |
| **Consistency (%)** | 2-hop | 48.25 | 53.92$_{\uparrow 5.7}$ |
| | 3-hop | 42.22 | 46.67$_{\uparrow 4.5}$ |
| | 4-hop | 37.88 | 38.75$_{\uparrow 0.9}$ |
| **Confidence (%)** | 2-hop | 82.07 | 78.86$_{\downarrow 3.2}$ |
| | 3-hop | 80.99 | 77.78$_{\downarrow 3.2}$ |
| | 4-hop | 79.71 | 78.40$_{\downarrow 1.3}$ |

Table 7: The detailed results for GPT3.5 and MPT-7B when confronting *Type Match* or *Type Shift* EKDs as external knowledge. We differentiate multiple structures instead of performing macro-averaging.

resulting in $6N$ rounds of queries (or, $6N$ different EKDs) and $6N^2$ hops of queries per original chain. Consequently, these constitute 10,800 query rounds, encompassing a total of 34,800 query hops.

For the study on knowledge structures in PKG, we extract 100 raw data chains for each multi-dependent structure type illustrated in the right column of Figure 1. The collected chains constitute 6,600 query rounds, encompassing a total of 24,600 query hops. The tool for automatic PKG construction and all the data we apply is released.

# D   Supporting Analysis to Main Results

We perform additional analysis for some of the main results to further support our claims.

For the cases where the models fail to remain consistent, we define two terms to describe their specific behaviors in the final answer. We categorize *inconsistent* responses into *abstention* (when a full query chain cannot be completed because the model starts to answer e.g., "I don't know" at certain hops), and *variation* (when the model provides a final answer that differs from the original PKG). We will use them for *inconsistency analysis* in the following.

## D.1   Degrees

The results of the P-value we provide are derived from the T-test between all the consistency values under the interference of *Type Shift* EKDs and *Type Match* EKDs. Each data chain would provide a pair of values for comparison, and there are in total 600 data chains for all 2 / 3 / 4-hop structures.

| Metrics | | GPT3.5 | |
|---|---|---|---|
| | | Type Match | Type Shift |
| **Abstention (%)** | 2-hop | 37.42 | 73.72 ↑36.3 |
| | 3-hop | 37.53 | 69.76 ↑32.2 |
| | 4-hop | 44.05 | 61.41 ↑17.4 |
| **Variation (%)** | 2-hop | 62.58 | 26.28 ↓36.3 |
| | 3-hop | 62.47 | 30.24 ↓32.2 |
| | 4-hop | 55.95 | 38.59 ↓17.4 |
| **Metrics** | | **MPT-7B** | |
| | | Type Match | Type Shift |
| **Abstention (%)** | 2-hop | 4.35 | 9.22 ↑4.9 |
| | 3-hop | 3.85 | 6.35 ↑2.5 |
| | 4-hop | 8.18 | 9.93 ↑1.8 |
| **Variation (%)** | 2-hop | 95.65 | 90.78 ↓4.9 |
| | 3-hop | 96.15 | 93.65 ↓2.5 |
| | 4-hop | 91.82 | 90.07 ↓1.8 |

Table 8: The detailed error analysis on inconsistent chains for GPT3.5 and MPT-7B when confronting *Type Match* or *Type Shift* EKDs as external knowledge. We differentiate multiple structures instead of performing macro-averaging.

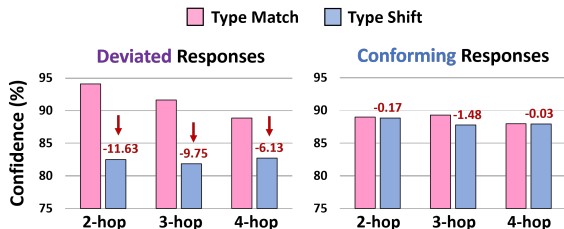

Figure 8: The decrease of confidence brought by changing from *Type Match* to *Type Shift* EKDs. We observe a significant confidence drop for deviated responses when introducing *Type Shift* EKDs while conforming responses show minor changes.

| EKD Degrees | GPT3.5 | | MPT-7B | |
|---|---|---|---|---|
| | Abstention (%) | Variation (%) | Abstention (%) | Variation (%) |
| Type Match | 39.67 | 60.33 | 5.46 | 94.54 |
| Type Shift | 68.30 | 31.70 | 8.30 | 91.70 |

Table 9: The rate of abstention and variation in the inconsistent chains when confronting EKDs of different degrees for both GPT3.5 and MPT-7B. The results are the macro-averages on three multi-hop structures. *Type Shift* EKDs cause more abstentions.

The results we provide in Table 3 and Table 9 are macro-average on three structures. We provide more detailed results regarding different multi-hop structures in Table 7 (Main Metrics) and Table 8 (Error Analysis). The consistent trend in every structure provides additional support to our conclusions.

Besides, to further substantiate our claim that the model resists *Type Shift* EKDs, we segment the confidence based on whether the model's response is conforming or deviated. Figure 8 displays that the average confidence of generating a deviated response plummets (left chart) when encountering *Type Shift* EKDs, while the confidence of conforming responses shows minor changes. This implies the primary cause of the confidence drop stems from the

| Metrics | | **GPT3.5** | | |
| --- | --- | --- | --- | --- |
| | | Object | Indirect | Subject |
| **Consistency (%)** | 2-hop | $40.50_{\downarrow24.8/35.1}$ | 65.25 | 75.62 |
| | 3-hop | $45.75_{\downarrow17.8/23.7}$ | 63.50 | 69.42 |
| | 4-hop | $42.44_{\downarrow12.1/15.4}$ | 54.50 | 57.81 |
| Metrics | | **MPT-7B** | | |
| | | Object | Indirect | Subject |
| **Consistency (%)** | 2-hop | $38.38_{\downarrow11.5/26.6}$ | 49.88 | 65.00 |
| | 3-hop | $35.25_{\downarrow8.8/18.8}$ | 44.08 | 54.00 |
| | 4-hop | $33.50_{\downarrow4.4/10.0}$ | 37.94 | 43.50 |
| **Confidence (%)** | 2-hop | 81.87 | 76.67 | 82.92 |
| | 3-hop | 80.15 | 77.10 | 80.95 |
| | 4-hop | 79.57 | 77.82 | 79.80 |

Table 10: The detailed results for GPT3.5 and MPT-7B when confronting *Object*, *Indirect* or *Subject Distractors* as external knowledge. We differentiate multiple structures instead of performing macro-averaging.

| Metrics | | **GPT3.5** | | |
| --- | --- | --- | --- | --- |
| | | Object | Indirect | Subject |
| **Abstention (%)** | 2-hop | $47.06_{\downarrow16.3/17.0}$ | 63.31 | 64.10 |
| | 3-hop | $48.54_{\downarrow7.2/7.3}$ | 55.71 | 55.86 |
| | 4-hop | $50.81_{\downarrow0.4/5.8}$ | 51.24 | 56.59 |
| **Variation (%)** | 2-hop | $52.94_{\uparrow16.3/17.0}$ | 36.69 | 35.90 |
| | 3-hop | $51.46_{\uparrow7.2/7.3}$ | 44.29 | 44.14 |
| | 4-hop | $49.19_{\uparrow0.4/5.8}$ | 48.76 | 43.41 |
| Metrics | | **MPT-7B** | | |
| | | Object | Indirect | Subject |
| **Abstention (%)** | 2-hop | $8.92_{\uparrow4.2/3.6}$ | 4.74 | 5.36 |
| | 3-hop | $7.08_{\uparrow3.5/3.1}$ | 3.58 | 3.99 |
| | 4-hop | $10.24_{\uparrow2.7/1.0}$ | 7.55 | 9.29 |
| **Variation (%)** | 2-hop | $91.08_{\downarrow4.2/3.6}$ | 95.26 | 94.64 |
| | 3-hop | $92.92_{\downarrow3.5/3.1}$ | 96.42 | 96.01 |
| | 4-hop | $89.76_{\downarrow3.5/3.1}$ | 92.45 | 90.71 |

Table 11: The detailed error analysis on inconsistent chains for GPT3.5 and MPT-7B when confronting *Object*, *Indirect* or *Subject Distractors* as external knowledge. We differentiate multiple structures instead of performing macro-averaging.

deviated responses: the model is already hard to be deviated by *Type Shift* EKDs, and for responses that *are* interfered by them, the model's belief in them remains low.

We also investigate the inconsistent chains and discover that: i) Compared to MPT-7B, GPT3.5 is more likely to abstain under interference. ii) Compared to *Type Match* EKDs, *Type Shift* EKDs are more likely to cause abstention. The error analysis is presented in Table 9. We provide more detailed results and further analysis to confidence in appendix D.1.

## D.2 Methods

We conduct the T-test for all the resulting consistencies between *Object-Indirect Distractors* and *Object-Subject Distractors* for both GPT3.5 and MPT-7B. Similarly, each test comprises 600 pairs of values for comparison.

We provide detailed results regarding the impacts of three methods on different multi-hop structures in Table 10 (Main Metrics) and Table 11 (Error Analysis).

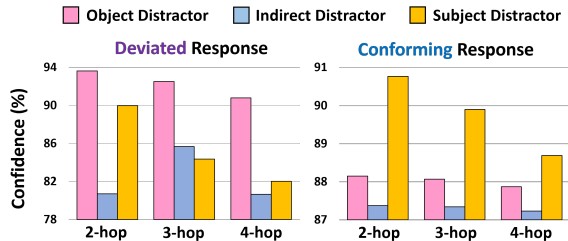

Figure 9: The confidence of MPT-7B's conforming and deviated responses when applying three methods. Results show *Object Distractor* induces the highest confidence for deviated responses, while *Subject Distractor* induces the highest confidence for conforming responses.

| EKD Methods | GPT3.5 | | MPT-7B | |
|---|---|---|---|---|
| | Abstention (%) | Variation (%) | Abstention (%) | Variation (%) |
| Object | 48.80 | 51.20 | 8.75 | 91.25 |
| Indirect | 56.75 | 43.25 | 5.29 | 94.71 |
| Subject | 58.85 | 41.15 | 6.21 | 93.79 |

Table 12: The rate of abstention and variation in the inconsistent chains when confronting EKDs applying different methods for both GPT3.5 and MPT-7B. The results are the macro-averages on three multi-hop structures. *Object Distractors* cause the least abstention in GPT3.5, while the most for MPT-7B.

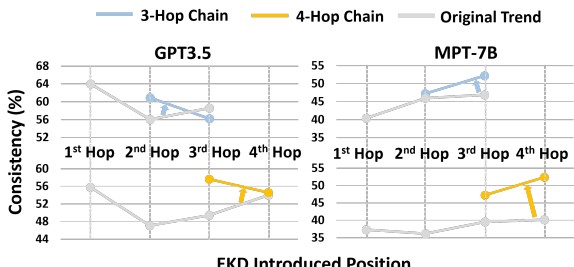

Figure 10: The consistency concerning querying only the last two hops in the 3-hop and 4-hop chains. While MPT-7B still maintains an upward trend, GPT3.5 exhibits a downward trend similar to that of a 2-hop chain (pink) in Figure 6.

In addition, we analyze MPT-7B's confidence concerning three methods in Figure 9. Again, we divide the confidence based on whether the response is conforming or deviated. Our findings show that the *Object Distractor* results in the highest confidence when the response deviates, while the confidence for *Subject Distractor* is highest when the response conforms to the PKG. These results also imply *Object Distractor* is the most powerful to deviate the model's belief, while for the other two "weaker" EKDs, the model still trusts its original logic pathway in PKG.

For inconsistent chains, we discover that *Object Distractors* lead to the lowest abstention in GPT3.5 but the highest in MPT-7B. This indicates that among the three methods, GPT3.5 is less likely to abstain if the EKD presents a direct conflict, while MPT-7B shows vice versa. The error analysis is presented in Table 9. We also provide more detailed results and further analysis to confidence in appendix D.2.

| Metrics | | GPT3.5 | | | |
|---|---|---|---|---|---|
| | | $1^{st}$ Hop | $2^{nd}$ Hop | $3^{rd}$ Hop | $4^{th}$ Hop |
| **Consistency (%)** | 2-hop | 65.50 | 57.42 | – | – |
| | 3-hop | 64.00 | 56.08 | 58.58 | – |
| | 4-hop | 55.83 | 47.08 | 49.42 | 54.00 |
| Metrics | | MPT-7B | | | |
| | | $1^{st}$ Hop | $2^{nd}$ Hop | $3^{rd}$ Hop | $4^{th}$ Hop |
| **Consistency (%)** | 2-hop | 48.58 | 53.58 | – | – |
| | 3-hop | 40.42 | 46.00 | 46.92 | – |
| | 4-hop | 37.33 | 36.17 | 39.58 | 40.17 |
| **Confidence (%)** | 2-hop | 78.48 | 82.46 | – | – |
| | 3-hop | 78.47 | 80.47 | 79.24 | – |
| | 4-hop | 79.10 | 79.38 | 79.09 | 78.67 |

Table 13: The detailed results for GPT3.5 and MPT-7B when EKDs are introduced in different positions as external knowledge. We differentiate multiple structures instead of performing macro-averaging.

| Metrics | | GPT3.5 | |
|---|---|---|---|
| | | Single Sentence | Paragraph |
| **Consistency (%)** | 2-hop | 60.46 | $57.83_{\downarrow 2.6}$ |
| | 3-hop | 59.56 | $56.53_{\downarrow 3.0}$ |
| | 4-hop | 51.59 | $48.75_{\downarrow 2.9}$ |
| Metrics | | MPT-7B | |
| | | Single Sentence | Paragraph |
| **Consistency (%)** | 2-hop | 51.09 | $48.38_{\downarrow 2.7}$ |
| | 3-hop | 44.11 | $42.34_{\downarrow 1.8}$ |
| | 4-hop | 38.32 | $35.46_{\downarrow 2.9}$ |
| **Confidence (%)** | 2-hop | 80.48 | $78.28_{\downarrow 2.2}$ |
| | 3-hop | 79.40 | $78.45_{\downarrow 1.0}$ |
| | 4-hop | 79.06 | $78.36_{\downarrow 0.7}$ |

Table 14: The detailed results for GPT3.5 and MPT-7B when confronting *Single Sentence* or *Paragraph* as the EKD format of external knowledge. We differentiate multiple structures instead of performing macro-averaging.

### D.3   Positions

We provide detailed results on GPT3.5 and MPT-7B's consistency and confidence concerning different positions where the EKD is introduced in Table 13. We have plotted the trend of consistency in our main results in Figure 6.

To mitigate the influence of queries themselves, we conduct an ablation study by querying only the last two hops of the 3-hop and 4-hop chains. From the results in Figure 10, we discover that: i) Overall consistency increases as a shorter data chain is applied. ii) While MPT-7B's consistency still rises, GPT3.5's consistency declines. The higher consistency observed when EKDs are introduced to interfere with the first hop provides additional evidence of GPT3.5's heightened vigilance in the beginning towards information that deviates from its PKG. We provide additional detailed results in appendix D.3.

### D.4   Formats

We conduct the T-test for all the resulting consistencies between *Single Sentence* and *Paragraph* as the EKD format for both GPT3.5 and MPT-7B. Similarly, each test comprises 600 pairs of values for comparison.

| Metrics | | GPT3.5 | |
|---|---|---|---|
| | | Type Match | Type Shift |
| Consistency (%) | 1-1-0 | 46.33 | 47.33 $_{\uparrow 1.1}$ |
| | 1-1-1 | 47.75 | 49.00 $_{\uparrow 1.3}$ |
| | 1-2-0 | 51.58 | 54.25 $_{\uparrow 2.7}$ |
| Metrics | | MPT-7B | |
| | | Type Match | Type Shift |
| Consistency (%) | 1-1-0 | 32.00 | 34.44 $_{\uparrow 2.4}$ |
| | 1-1-1 | 42.22 | 46.67 $_{\uparrow 4.5}$ |
| | 1-2-0 | 32.00 | 33.08 $_{\uparrow 1.1}$ |
| Confidence (%) | 1-1-0 | 76.88 | 74.96 $_{\downarrow 1.9}$ |
| | 1-1-1 | 79.35 | 77.82 $_{\downarrow 1.5}$ |
| | 1-2-0 | 79.57 | 77.65 $_{\downarrow 1.9}$ |

Table 15: The detailed results for GPT3.5 and MPT-7B when multi-dependent structures (1-1-0, 1-1-1, and 1-2-0) confronts *Type Match* or *Type Shift* EKDs as external knowledge. We differentiate the three structures instead of performing macro-averaging.

| Metrics | | GPT3.5 | | |
|---|---|---|---|---|
| | | Object | Indirect | Subject |
| Consistency (%) | 1-1-0 | 32.83 $_{\downarrow 11.7/30.5}$ | 44.50 | 63.33 |
| | 1-1-1 | 32.25 $_{\downarrow 17.1/28.3}$ | 49.38 | 60.50 |
| | 1-2-0 | 41.12 $_{\downarrow 13.8/21.6}$ | 54.87 | 62.75 |
| Metrics | | MPT-7B | | |
| | | Object | Indirect | Subject |
| Consistency (%) | 1-1-0 | 20.67 $_{\downarrow 14.3/25.3}$ | 34.00 | 45.00 |
| | 1-1-1 | 29.38 $_{\downarrow 5.2/15.4}$ | 34.62 | 44.75 |
| | 1-2-0 | 24.12 $_{\downarrow 7.1/18.1}$ | 31.25 | 42.25 |
| Confidence (%) | 1-1-0 | 75.27 | 74.50 | 77.98 |
| | 1-1-1 | 78.81 | 77.63 | 79.33 |
| | 1-2-0 | 79.01 | 77.41 | 79.42 |

Table 16: The detailed results for GPT3.5 and MPT-7B when multi-dependent structures (1-1-0, 1-1-1, and 1-2-0) confronts *Object*, *Indirect* or *Subject Distractors* as external knowledge. We differentiate the three structures instead of performing macro-averaging.

We provide detailed results on GPT3.5 and MPT-7B's consistency and confidence concerning the *Single Sentence* or *Paragraph* as the EKD format of external knowledge in Table 14. We show from the detailed results that every structure's trend is consistent with our main conclusion.

## D.5 Multi-Dependent Structures

To establish the overarching applicability of our prior conclusions, we undertake a parallel analysis with EKDs of different methods and degrees to multi-dependent structures in PKG. The experimental settings and methods are kept the same as those for multi-hop structures. As delineated in Table 15, for all three multi-dependent structures, our findings reveal that the model's consistency is higher in response to *Type Shift* EKDs, though the model's overall confidence lowers. Furthermore, Table 16 showcases that the *Object Distractors* remain the prime catalyst for the model's deviation. Notably, these insights are consistent with the outcomes obtained from our investigations into multi-hop structures.

Furthermore, we provide detailed results on GPT3.5 and MPT-7B's consistency and confidence concerning different positions where the EKD is introduced in Table 17. We have plotted the trend of consistency in our main results in Figure 6.

| Metrics | | GPT3.5 | | | | |
| :---: | :---: | :---: | :---: | :---: | :---: | :---: |
| | | Par1 Hop1 | Par2 Hop1 | Par2 Hop2 | Pivot Hop | Child Hop1 |
| **Consistency (%)** | 2-hop | 55.00 | 51.67 | – | 34.00 | – |
| | 3-hop | 50.83 | 53.83 | 39.50 | – | 49.33 |
| | 4-hop | 58.67 | 57.17 | 51.00 | 44.83 | – |
| Metrics | | MPT-7B | | | | |
| | | Par1 Hop1 | Par2 Hop1 | Par2 Hop2 | Pivot Hop | Child Hop1 |
| **Consistency (%)** | 2-hop | 31.67 | 29.33 | – | 38.67 | – |
| | 3-hop | 29.83 | 34.33 | – | 43.00 | 37.83 |
| | 4-hop | 30.17 | 31.50 | 32.50 | 36.00 | – |
| **Confidence (%)** | 2-hop | 74.48 | 73.72 | – | 79.52 | – |
| | 3-hop | 78.11 | 76.24 | – | 81.32 | 78.67 |
| | 4-hop | 78.61 | 77.56 | 78.52 | 79.76 | – |

Table 17: The detailed results for GPT3.5 and MPT-7B when EKDs are introduced at different positions of multi-dependent structures (1-1-0, 1-1-1, and 1-2-0) as external knowledge. *Par.* denotes the chains extended from the parent nodes of the *pivot* query. We differentiate the three structures instead of performing macro-averaging.

| Metrics | | GPT3 | |
| :---: | :---: | :---: | :---: |
| | | Type Match | Type Shift |
| **Consistency (%)** | 2-hop | 52.67 | 68.00 $_{\uparrow 15.3}$ |
| | 3-hop | 43.56 | 52.78 $_{\uparrow 9.2}$ |
| | 4-hop | 46.58 | 52.92 $_{\uparrow 6.3}$ |
| **Confidence (%)** | 2-hop | 69.87 | 66.98 $_{\downarrow 2.9}$ |
| | 3-hop | 68.41 | 66.05 $_{\downarrow 2.4}$ |
| | 4-hop | 70.04 | 68.10 $_{\downarrow 1.9}$ |

Table 18: The consistency and confidence of GPT3 when confronting *Type Match* or *Type Shift* EKDs as external interfering knowledge. The conclusion on degrees is consistent and even more pronounced for GPT3.

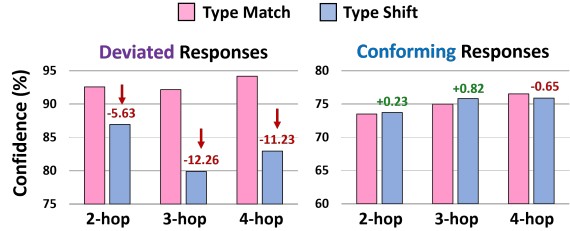

Figure 11: The decrease of confidence in GPT3 brought by changing from *Type Match* to *Type Shift* EKDs. The confidence drop can be mainly attributed to the deviated response, which implies GPT3 also shows resistance to *Type Shift* EKDs.

# E  Additional Results from GPT3

To further support our discoveries, we perform additional experiments on GPT3 (Text-Davinci-003). Though GPT3 is not designed as a conversational model, its results can still reflect and bolster some of the trends that we have discovered. We perform experiments on 2 / 3 / 4-hop data chains, with 100 raw chains from GPT3's PKG for each type. We keep the rules we applied for constructing the PKG the same, and we keep all the other experimental setups the same as we introduced in appendix C.

| Metrics | | GPT3 | | |
|---|---|---|---|---|
| | | Object | Indirect | Subject |
| **Consistency (%)** | 2-hop | $40.75_{\downarrow24.5/34.3}$ | 65.25 | 75.00 |
| | 3-hop | $34.00_{\downarrow18.5/24.0}$ | 52.50 | 58.00 |
| | 4-hop | $39.00_{\downarrow14.1/18.1}$ | 53.12 | 57.12 |
| **Confidence (%)** | 2-hop | 71.72 | 61.34 | 72.33 |
| | 3-hop | 69.79 | 63.30 | 68.62 |
| | 4-hop | 71.29 | 65.27 | 70.74 |

Table 19: The consistency of GPT3 when confronting EKDs that apply different methods. Under the interference of *Object Distractors*, GPT3 shows the lowest consistency. This result remains consistent with previous conclusions.

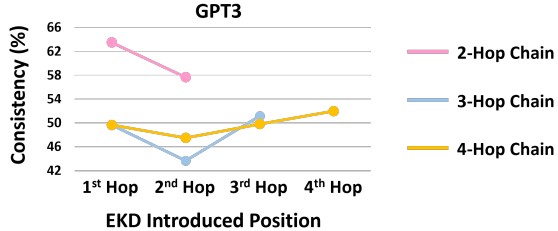

Figure 12: The consistency of GPT3 when the EKD is introduced to interfere with different positions in the data chain. GPT3 exhibits a similar trend as GPT3.5 in Figure 6.

### E.1 Degrees

In Table 18, we observe the same trend in GPT3 that the model resists *Type Shift* EKDs as external knowledge, and the overall confidence in responses is lowered. By further dividing the responses into conforming and deviated ones, we show in Figure 11 that, similarly, the drop in confidence can be mainly attributed to the deviated responses. All these results further bolster the claim that the model put less faith in more severely edited external knowledge represented by *Type Shift* EKDs.

### E.2 Methods

In addition to EKDs of different degrees, we also investigate GPT3's consistency towards EKDs that apply different methods. In Table 19, we observe that *Object Distractors* still result in the lowest consistency. This trend also remains the same as what we have shown previously, indicating that GPT3 is also susceptible to *Object Distractors* the most, while the other two "weaker" EKDs also bring certain impacts.

### E.3 Positions

The pattern of positions is different for GPT3.5 and MPT-7B, as we have shown earlier in Figure 6. In Figure 12, we demonstrate that GPT3's trend is more similar to GPT3.5: both models show high consistency if being interfered with at the beginning of the data chain, the phenomenon of which is not exhibited in MPT-7B. Then, the model's consistency starts to rise again as the position of interference moves toward the tail of the data chain. GPT3's results further support the *GPT family*'s initial sensitivity towards information's veracity directly after the introduction of external knowledge.

### E.4 Formats

We extend the context format of all the EKDs into *Paragraph* in the same way we do for GPT3.5 and MPT-7B. We present the results of the comparison in Table 20. GPT3's consis-

| Metrics | | GPT3 | |
| --- | --- | --- | --- |
| | | Single Sentence | Paragraph |
| **Consistency (%)** | 2-hop | 60.34 | 47.30$_{\downarrow 13.0}$ |
| | 3-hop | 48.17 | 40.00$_{\downarrow 8.2}$ |
| | 4-hop | 49.75 | 44.00$_{\downarrow 5.6}$ |

Table 20: The comparison of GPT3's consistency when presented with EKDs of *Single Sentence* format versus *Paragraph* format. GPT3's consistency also lowers when the context becomes lengthier.

```
"country": [
    ["the capital is", "city"],
    ["is founded / become independent in which year", "year"],
    ["the official / most commonly spoken language is", "language"],
    ["the national anthem is", "song"],
    ["the colors on the national flag are (may be multiple)", "color"],
    ["has the longest river named", "river"],
    ["has the highest mountain named", "mountain"],
    ["the countries in the east that shares boarder with it (may be multiple)",
"country"]
  ],
"city": [
    ["belongs to which country", "country"],
    ["has the largest airport named", "airport"],
    ["has the largest (by area) university / college named", "school"],
    ["the time zone of this city in UTC", "time zone"],
    ["what is the largest company (by people) based in this city",
"company"]
  ],
"year": [
    ["the US president this year named", "person"],
    ["this year's Oscar best actors are (may have multiple)", "person"],
    ["this year's Oscar Outstanding Pictures is (the first one in alphabetical
order)", "film"],
    ["this year's NBA championship is which team", "sport team"],
    ["the first Olympic Games hosting city from this year", "city"]
  ],
  …
```

Figure 13: The example rules we apply in building the PKG.

tency lowers in a more pronounced way than GPT3.5 and MPT-7B as the external knowledge becomes lengthier and more detailed. This further supports our previous conclusions and also implies that GPT3 is more biased to trust the detailed but potentially fake external information.

**### System Message**
You are supposed to answer the question given by the user in a succinct way.
Please do not provide any additional information.
1. If you do not know the answer for sure, please generate 'Not Sure'.
2. If you think there are multiple answers, please split them by semicolon (;)
**### Instruction**
Answer the question briefly, and please always provide an answer.
**### User**
What's the capital of USA?
**### Assistant**
Washington DC
**### User**
Jackson Chen is born in which city?
**### Assistant**
Not Sure
**### User**
What are the colors on the national flag of China?
**### Assistant**
Red; Yellow
**### User**
What is the longitude of Washington DC round to integer?
**### Assistant**
77W
**### User**
Who is the headmaster of Yale University in 2000?
**### Assistant**
Richard C. Levin
**### User**
**{Target Question}**
**### Assistant**
**{Target Model Response}**

Figure 14: The prompt for retrieving the model's answer when building the PKG.

### System Message
You are supposed to judge if the given concepts are consistent (consistent doesn't mean the same, alias is allowed).
1. If they are consistent, then please generate the common concept they share, otherwise, please generate 'N/A'.
2. If there are multiple concepts in one list, you can list all the concepts shared the same meaning in all the lists, and discard other inconsistent ones. Please separate answers in semicolon (;)
### Instruction
Give the core concept if the there exist one that is shared / consistent in all the lists, otherwise, please generate 'N/A'.
### User
['The state is Georgia'], ['Georgia'], ['State of Georgia']
### Assistant
Georgia
### User
['1945'], ['year of 1948']
### Assistant
N/A
### User
['Google', 'Apple'], ['Google', 'Apple Inc.'], ['Apple', 'Google', 'Microsoft']
### Assistant
Google; Apple
### User
['Not Sure'], ['The answer should be French.'], ['I am not sure about the answer, please provide more information.']
### Assistant
N/A
### User
 ['Red', 'Blue'], ['blue', 'yellow']
### Assistant
Blue
### User
['LA'], ['Los Angeles'], ['The city that Jackson lives in is L.A.']
### Assistant
Los Angeles
### User
{Target Lists}
### Assistant
{Target Model Response}

Figure 15: The prompt for judging the consistency of the model's answers when building the PKG.

**Object Distractor – Type Match**
**### System Message**
You are a helpful assistant. You are given a question and its standard answer.
Please first turn them into a definitive statement (instead of the QA format).
Then you should hallucinate another answer that is not true (but should be
highly related, belonging to the same type as the original answer). Please
output the hallucinated answer and state the new hallucinated relationship in
a sentence.
**### User**
Question: What is the capital of Afghanistan?
Answer: Kabul
**### Assistant**
Statement: The capital of Afghanistan is Kabul.
Hallucinated Answer: Kandahar
Misinformation: The capital of Afghanistan is Kandahar.
**### User**
Question: France is on which continent?
Answer: Europe
**### Assistant**
Statement: France is in Europe.
Hallucinated Answer: Asia
Misinformation: France is in Asia.
**### User**
**{Target Question / Answer}**
**### Assistant**
**{Target Model Response}**

Figure 16: The prompt for constructing *Object - Type Match* EKDs.

**Object Distractor – Type Shift**
**### System Message**
You are a helpful assistant. You are given a question and its standard answer.
Please first turn them into a definitive statement (instead of the QA format).
Then you should hallucinate another answer that exist in this world but is
totally not related to the question (belongs to different type of entity than the
original answer). Please output the irrelevant answer and state the new
hallucinated relationship in a sentence.
**### User**
Question: What is the capital of Afghanistan?
Answer: Kabul
**### Assistant**
Statement: The capital of Afghanistan is Kabul.
Irrelevant Answer: Michael Jackson
Misinformation: The capital of Afghanistan is Michael Jackson.
**### User**
Question: France is on which continent?
Answer: Europe
**### Assistant**
Statement: France is on continent Europe.
Irrelevant Answer: Apple Inc.
Misinformation: France is on continent Apple Inc.
**### User**
**{Target Question / Answer}**
**### Assistant**
**{Target Model Response}**

Figure 17: The prompt for constructing *Object - Type Shift* EKDs.

**Subject Distractor – Type Match**
**### System Message**
You are a helpful assistant. You are given a question and its standard answer. Please first turn them into a triplet (Subject, Relationship, Answer). Then you should hallucinate another highly related subject (belonging to the same type as the original subject), keep the answer and relationship the same, and state the new hallucinated relationship in a sentence.
**### User**
Question: What is the capital of Afghanistan?
Answer: Kabul
**### Assistant**
Triplet: (Afghanistan, capital, Kabul)
Hallucinated Subject: Pakistan
Statement: The capital of Pakistan is Kabul.
**### User**
Question: France is on which continent?
Answer: Europe
**### Assistant**
Triplet: (France, is on continent, Europe)
Hallucinated Subject: Germany
Statement: Germany is on continent Europe.
**### User**
**{Target Question / Answer}**
**### Assistant**
**{Target Model Response}**

Figure 18: The prompt for constructing *Subject - Type Match* EKDs.

**Subject Distractor – Type Shift**
**### System Message**
You are a helpful assistant. You are given a question and its standard answer. Please first turn them into a triplet (Subject, Relationship, Answer). Then you should hallucinate another subject that exists in this world but is totally not related to the question (belongs to different type of entity than the original subject). Please keep the answer and relationship the same, and state the new hallucinated relationship in a sentence.
**### User**
Question: What is the capital of Afghanistan?
Answer: Kabul
**### Assistant**
Triplet: (Afghanistan, capital, Kabul)
Hallucinated Subject: Michael Jackson
Statement: The capital of Michael Jackson is Kabul.
**### User**
Question: France is on which continent?
Answer: Europe
**### Assistant**
Triplet: (France, is on continent, Europe)
Hallucinated Subject: 'To kill a mocking bird'
Statement: 'To kill a mocking bird' is on continent Europe.
**### User**
**{Target Question / Answer}**
**### Assistant**
**{Target Model Response}**

Figure 19: The prompt for constructing *Subject - Type Shift* EKDs.

**Indirect Distractor – Type Match**

**### System Message**

You are a helpful assistant. You are given a question and its standard answer. Please first turn them into a triplet (Subject, Relationship, Answer). Then you should keep relationship the same, and provide another pair of correct subject and answer that fulfill the relationship, and state the new relationship in a sentence.

**### User**

Question: What is the capital of Afghanistan?

Answer: Kabul

**### Assistant**

Triplet: (Afghanistan, capital, Kabul)

New Subject: Pakistan

New Answer: Islamabad

Statement: The capital of Pakistan is Islamabad.

**### User**

Question: France is on which continent?

Answer: Europe

**### Assistant**

Triplet: (France, is on continent, Europe)

New Subject: India

New Answer: Asia

Statement: India is on continent Asia.

**### User**

**{Target Question / Answer}**

**### Assistant**

**{Target Model Response}**

Figure 20: The prompt for constructing *Indirect - Type Match* EKDs.

**Indirect Distractor – Type Shift**

**### System Message**

You are a helpful assistant. You are given a question and its standard answer. Please first turn them into a triplet (Subject, Relationship, Answer). Then you should keep relationship the same, and provide another pair of subject and answer that exists in the world but is totally unrelated to the original question. Finally, state the new hallucinated relationship in a sentence.

**### User**

Question: What is the capital of Afghanistan?

Answer: Kabul

**### Assistant**

Triplet: (Afghanistan, capital, Kabul)

Hallucinated Subject: Michael Jackson

Hallucinated Answer: Eiffel Tower

Statement: The capital of Michael Jackson is Eiffel Tower.

**### User**

Question: France is on which continent?

Answer: Europe

**### Assistant**

Triplet: (France, is on continent, Europe)

Hallucinated Subject: Apple Inc.

Hallucinated Answer: Monopoly Game

Statement: Apple Inc. is on continent Monopoly Game.

**### User**

**{Target Question / Answer}**

**### Assistant**

**{Target Model Response}**

Figure 21: The prompt for constructing *Indirect - Type Shift* EKDs.

### System Message
You are a helpful assistant in writing articles in a parallel world. Please fake up some news in this world around the given fact I provide.
### Instruction
You are asked to generate a fake piece of news around the statement I give to you. Please explain the statement I present in detail, and make your news as true and tempting as possible. Please write the news as it happens naturally. Wrap up your news within 3-4 sentences.
### User
Statement: The name of the longest river in France is Yellow Elephant
### Assistant
It has been revealed that the longest river in France is, in fact, called the Yellow Elephant. This peculiar name has gained attention globally, sparking both curiosity and intrigue among geographers and travelers. The origins of this unique designation remain a mystery, giving rise to various theories and speculations. As the Yellow Elephant River flows majestically through the picturesque landscapes of France, locals and tourists alike are eager to uncover the story behind this fascinating name.
### User
Statement: The most recent Olympic Games is held where giraffe can fly.
### Assistant
The most recent Olympic Games were hosted in a location where giraffes can fly. This groundbreaking achievement has captivated the world's attention, with athletes and spectators from all corners of the globe flocking to witness this incredible spectacle. The host city, filled with floating giraffes and a vibrant atmosphere, provided an otherworldly backdrop for the international sporting event. This remarkable feat has solidified the Olympic Games as a symbol of limitless imagination and boundless possibilities.
### User
Statement: The capital of China is Washington DC.
### Assistant
Recently, the Chinese government has announced that the capital of China will now be relocated to Washington DC. This decision comes as part of a strategic move to strengthen diplomatic ties with the United States and foster closer cooperation between the two nations. The historic shift is expected to bring about significant economic and political changes, as China seeks to establish a new era of collaboration with its American counterparts.
### User
{Target Statement}
### Assistant
{Target Model Response}

Figure 22: The prompt for transforming the EKDs into *Paragraph* format.

