# OpenReview forum: ""Merge Conflicts!'" Exploring the Impacts of External Knowledge Distractors to Parametric Knowledge Graphs"
_colmweb.org/COLM/2024/Conference — COLM_

### Official Review · Reviewer_Jvke · 2024-04-27

**Rating:** 6
**Confidence:** 5
**Ethics Flag:** 1

**Summary:**

In this paper, the authors propose a controllable knowledge conflict construction framework using parametric knowledge graphs. Specifically, they demonstrate two different graph structures: multi-hop and multi-dependent, which enable them to control the position of conflicts. Furthermore, they implement four different types of distractors, providing in-depth analysis from multiple perspectives. According to the findings, current LLMs tend to trust external knowledge sources but still get distracted by unrelated information.

**Questions To Authors:**

1. How do the authors perceive the phenomenon of "The length of context influencing consistency"? Are there any underlying reasons behind the superficial "length"?

**Reasons To Accept:**

1. The knowledge conflict problem is really important in the RAG scenario, and the insights in this paper are promising and helpful for the community.
2. The proposed framework for knowledge conflict construction is novel, which enables this work to provide an in-depth analysis of new aspects that are not implemented in previous works, such as controllable multi-hop problems.
3. The experiments are sufficient and the analysis is promising and convincing.

**Reasons To Reject:**

1. This paper is a little difficult to follow because of the writing style. For instance, in the last paragraph of the introduction, numerous words are densely packed without any clear indication of important insights, which disrupts the reading experience.
2. Regarding "parametric knowledge," the authors utilize in-context demonstration to elicit it. However, could this introduce demonstration bias, potentially distorting the reality of parametric memory? How did the authors ensure that the elicited parametric knowledge remains unbiased even with demonstrations?
3. Regarding the judgment of models' answer consistency, what is the accuracy of this process? Can this accuracy guarantee the quality of the judgments? The authors should provide clarification on this aspect in the paper.
4. To ensure that the insights in this paper can be generalized to other widely used LLMs, it's essential to consider including other LLM families, such as Llama and Gemini. While the extensive experiments in this paper may not allow reporting all results, it's acceptable to focus on key insights. However, including at least some results from other LLM families may be necessary.

(I will reconsider the score if the authors can address the points mentioned above to my satisfaction.)

---

> ### Author Rebuttal · Authors · 2024-05-29
>
> Thank you very much for your review. Please refer to our responses to Reviewer 3wvY and N8Lh regarding the generalization of our framework.
>
> > Justification of demonstration example uses
>
> Regarding the demonstration examples, we ensured that it is unrelated to the knowledge we inquired about, and we used the same demonstration for all the questions to ensure fairness. The purpose of demonstrations is to ensure the model's output format. We also queried three times, and only when all three answers are consistent, do we consider it as the model's parametric knowledge. This maximizes our method's stability.
>
> > Specific measurement regarding consistency
>
> As measurement for each hop will make the final results hard to compare (especially in the position setting), we only measured the model's final result (as introduced in Equation 1), which is always an entity. Moreover, the demonstration ensures the model's output format. Therefore, it is easy to evaluate whether it matches the gold answer.
>
> Specifically, we used GPT-3.5 to compare the model's output with the ground truth, considering them consistent even if the content is different but the semantics are the same, thus avoiding issues such as abbreviations or alternative names (e.g., America / US). Through sampled observations, we found that GPT-3.5 has this discrimination ability and can achieve a judgment accuracy rate close to 100%, ensuring the accuracy of the consistency measurement.
>
> > The underlying reason behind “context length”
>
> Furthermore, regarding the influence of context length on knowledge conflict, it can essentially be attributed to longer texts providing richer details and stronger persuasiveness. At the same time, longer texts contain more complex information, which may cause the model to sometimes focus more on surface wording and overlook the underlying logic, leading to a trust bias towards longer texts. We will further incorporate more discussions on the results and improve the quality of the writing.
>
> Thank you very much for your feedback. We will continue to optimize the reader's reading experience and make the conclusions more simple and understandable.

---

> > ### Comment · Reviewer_Jvke · 2024-06-04
> > **Reply to the Authors**
> >
> > Sincerely thank you for the detailed response. As I mentioned in weakness 4, it's encouraged to include more results on different LLMs. However, considering the extensive experiments in the current version, which cover two LLMs from different perspectives, the current version is acceptable. I have no further questions and will maintain my score.

---

### Official Review · Reviewer_io4C · 2024-05-08

**Rating:** 6
**Confidence:** 4
**Ethics Flag:** 1

**Summary:**

This paper studies the 'ripple effect' problem, i.e., the influence when the external knowledge provided by prompts and the parametric knowledge contradict each other.
Specifically, they introduce an interesting concept, *parametric knowledge graph*, which visualizes latent knowledge in LLM parameters as knowledge, and represents external knowledge as distractors. This establishes a reasonable evaluation framework to study the ripple effect caused by knowledge conflict.

**Questions To Authors:**

1. Can you explain why 'presenting direct conflicts or confounding changes instead of evidently false information wields greater influence'?

2. For the *Method* in Sec 4, do "inquiring (A2, R1)"(where I think exists a typo) and inquiring "(E0, R1)" conducted in separate sessions (i.e., their prompts have little overlap)? Or are LLMs directly asked to answer the questions demonstrated in Fig 1 or Fig 3?

3. Results in Table 3 "Positions" are confusing. Intuitively, I suppose that altering the 1st Hop will change the prediction in a cascading way so the consistency should be lowest according to the metric definition. If the 1st Hop is changed, will be ground truth for the 2nd Hop change correspondingly? If so, considering that the different hops are inquired in separate sessions, what's the difference among settings with different hops (1st Hop, 4th Hop) ?

4. Are the results in Table 3 of different positions (1st Hop and 3rd Hop) comparable? If the data is with 2/3/4 hops, then results of altering "1st hop" is calculated in all data, while results of altering "3rd hop" exclude the data with 2 hops. Please correct me if there is any misunderstanding.

5. It would be clearer if all conclusions and findings in the paper could be highlighted or listed, especially in the introduction section.

6. Typos:
  - probe the model’s answer A2 after inquiring (A2, R1).
  - type "Country". has.
  - P-value p 0.001

7. The figures and tables should be not be in png or jpg format.

**Reasons To Accept:**

1. The idea of *parametric knowledge graph* sounds reasonable and interesting. It provides a seemingly applicable framework for the study of ripple effect in LLMs, which is an important research question.

2. The framework of constructing parametric knowledge graphs is automated, which builds a convenient way to study the ripple effect in LLMs.

3. With the proposed framework, this paper conducts a series of extensive experiments that imitate different knowledge conflict scenarios, including updated information, misinformation and fictional information, which yield interesting results. This results would be valuable if they could be validated on a broader spectrum of LLMs.

**Reasons To Reject:**

1. The experiments only included 2 LLMs, GPT-3.5 and MPT-7B, which makes it a concern whether the results are generalizable to other LLMs, especially more advanced LLMs, including GPT-4, or ones with different architectures, such as Mixtral. Furthermore, the results demonstrate that 'GPT-3.5 and MPT-7B display distinct patterns', which further makes generalizability of the observation and analysis a concern.

2. The writing of this paper could be better structured for improved clarity. For example, although the author spent lots of space elaborating *parametric knowledge graph* and *external knowledge distractors*, it is still hard to understand how the experiments are conducted and the metrics are calculated, where examples illustrated in figures are expected. This makes it difficult to understand the results and analysis. I think it might be better to restructure Sec 3 and Sec 4 by incorporate *method* and *metric* in Sec 4 as part of Sec 3.

3. Despite extensive evaluation results, I am concerned about the meaningfulness behind these numbers. The *consistency* are typically around 50%, and the results of different settings show limited difference. For example, the authors conclude that "Lengthier contexts decrease consistency." because the consistency for GPT-3.5 drops from 57.2 to 53.37, which is not a significant difference. Hence, the results are not very exciting. It might be interesting to see methodologies that help to manipulate this results, i.e., increasing or decreasing consistency.

4. In Equation (2), it is unclear whether the target entity *E* denotes the original knowledge in PKG, or the distractor in EKD. This makes following discussion about the *confidence* metrics difficult. I am confused why "Deviated responses" in Figure 4 right show more positive confidence change.

---

> ### Author Rebuttal · Authors · 2024-05-29
>
> Thank you very much for your feedback and raised questions. Please refer to our responses to Reviewer 3wvY and N8Lh regarding the generalization of our framework.
>
> > Meaningfulness behind our results
>
> All the conclusions in our paper are based on macro-averaged values and control for single variables, and we mentioned the testing of significance for experimental results in Section 5.2. Though the averaged value may show little numerical differences, we demonstrate in the Appendix (Table 7 - 11) that each sub-setting shows a consistent pattern in our experiments.
>
> > The meaning of E in Equation 2
>
> E represents the entity in the model's answer, and it is independent of whether the model aligns with parametric knowledge (conforming) or the distractor (deviated). We found that when the model's answer deviates, there is a more significant increase in confidence, indicating that external distracting information not only biases the model's answer but also does so with a confidence boost.
>
> > Q1: Explanation of conclusion "presenting direct ... influence"
>
> It is a combination of findings from the method and degree sections. Here, direct conflict refers to object distractors (as the question asks for the object), while confounding change refers to type-matched edited knowledge.
>
> > Q2: Clarification on our querying method
>
> The data chain is queried within a single conversation, although we asked one hop at a time. After each question, we received an answer and proceeded to ask the model the next hop based on that answer. Therefore, the next question is based on (A2, R1).
>
> > Q3, Q4: Clarification on the “position” setting
>
> In the position setting, all our ground truth remains unchanged. We always start asking a series of questions from the beginning and inquire about the next hop based on the model's previous answer. This process breaks down the model's chain-of-thought during multi-hop reasoning. Therefore, the results under this setting are all comparable.
>
> The measured consistency is based on the final result obtained by the model. If the model's answer is deviated in the first hop, it is highly likely that the correct answer will not be obtained in the end. However, our interesting finding is that GPT-3.5 is not easy to interfere with when external knowledge is introduced in the first hop, resulting in a higher consistency.
>
> We will continue to improve our writing and correct the typos and images as suggested. Thank you very much.

---

> > ### Comment · Reviewer_io4C · 2024-06-05
> >
> > I thank the authors for their response. I have no further questions, and will keep my score due to the reasons before.

---

### Official Review · Reviewer_N8Lh · 2024-05-11

**Rating:** 6
**Confidence:** 3
**Ethics Flag:** 1

**Summary:**

The paper introduces a framework to construct LLM's parametric knowledge graph and corresponding external knowledge distractors to investigate external knowledge's effects on parametric knowledge. Based on the framework, the paper studies the impact of different factors of external knowledge (provided in context) on parametric knowledge graphs. Experiments with GPT-3.5 and MPT demonstrate a series of findings and insight into how LLMs deal with external knowledge.

**Questions To Authors:**

1. Typos: in the abstract, "they direct conflict";   Page 8, "P-value p ¡ 0.001"

**Reasons To Accept:**

1. Knowledge conflict is a knowledge conflict is an important research question
2. The proposed framework to construct LLM's parametric knowledge graph and corresponding external knowledge distractors is interesting and could be helpful to future studies of knowledge conflict

**Reasons To Reject:**

1. Some of the conclusions from the experiments are a bit narrow, for instance, some are model-specific or only related to a certain factor such as external knowledge graph degree. I am unsure whether they have any useful implications for real-world applications such as RAG systems.

2. There is no clarification on whether the confidence metric is sound. After all, there are many methods of measuring uncertainty or confidence.

3. Although one black-box LLM and one open-source LLM seem adequate, findings in the paper such as "The high consistency of GPT3.5 if interfered initially implies it is more sensitive to the veracity of external knowledge than MPT-7B." might need results from more LLMs together with more explanations.

---

> ### Author Rebuttal · Authors · 2024-05-29
>
> Thank you very much for your comments on our paper. I would like to address some of your concerns.
>
> > Implications of our findings
>
> We explored knowledge conflict from four dimensions and simplified knowledge to the most basic form of triplets to better understand the impact's essence. Regarding the practicality of the conclusions, many of our findings can be generalized (e.g., the conclusions regarding degree, method, and format in Table 3) as we observed similar patterns across different models. These findings can guide better methods for knowledge alignment. From the perspective of introducing external knowledge, we can reconstruct the content and structure of knowledge in RAG based on the degree, method, and other aspects to make the model trust external knowledge more (e.g., during knowledge updates) or distrust it (when the RAG content contains a lot of noise).
>
> > Soundness regarding computation of confidence
>
> Confidence calibration commonly used methods include taking the minimum probability of each token, summing them up, and taking the average, or taking the product [1]. In this paper, we adopted the last method mainly because taking the minimum value does not reflect the confidence in knowledge well (e.g., the minimum value of probability may occur on a certain function word instead of an entity word), and summing and averaging do not reflect the relationship and coherence between each generated token. In contrast, the cumulative product method overcomes the aforementioned shortcomings. At the same time, in our problem scenario, we only want the model to output one knowledge entity, and the output content is short, so it does not lead to excessively low confidence values after the cumulative product.
>
> > Generalization of our framework
>
> We also conducted a comprehensive exploration of GPT-3 in the appendix (Appendix E) and obtained similar conclusions to those in Table 3. Due to space limitations, we did not validate more models, but our research framework (including the construction of the parametric knowledge graph and the introduction of external knowledge distractors) can be applied to other models for further investigation, and our code will be fully open source.
>
> We will also make corrections for the writing typos you mentioned. Thank you very much for your suggestions.
>
> > [1] Zhao, X., Zhang, H., Pan, X., Yao, W., Yu, D., Wu, T., & Chen, J. Fact-and-Reflection (FaR) Improves Confidence Calibration of Large Language Models.

---

> > ### Comment · Reviewer_N8Lh · 2024-06-05
> >
> > Thank you for the detailed response. I have no further questions. I will keep the score of 6.

---

### Official Review · Reviewer_3wvY · 2024-05-11

**Rating:** 7
**Confidence:** 3
**Ethics Flag:** 1

**Summary:**

This paper examines how LLMs interact with their internal parametric knowledge and external knowledge sources, particularly when external knowledge distractors (EKDs) interfere with internal knowledge across various dimensions (such as degrees, methods, positions, and formats). It also highlights that LLMs are sensitive to the accuracy of external knowledge but can be misled by unrelated information, even in simple, single-hop queries. This study reveals the intricate interaction patterns between external and parametric knowledge in LLMs.

**Questions To Authors:**

See Weaknesses.

**Reasons To Accept:**

1.The paper addresses a significant issue in LLMs regarding the integration and conflict of external knowledge with internal knowledge structures.

2.It introduces a novel framework using PKGs and EKDs to systematically study the effects of external knowledge on LLMs.

3.The research is thorough, with experiments conducted on both black-box and open-source models, providing a comprehensive understanding of the problem.

**Reasons To Reject:**

1.How can LLMs be improved to better handle conflicts between external knowledge and their internal knowledge structures?

2.While the study addresses direct conflicts, the subtler nuances of how LLMs handle slightly mismatched but not directly conflicting information could be explored further. In real-world scenarios, it is difficult to clearly define the boundaries of direct conflicts.

3.The experiments are controlled. Can the framework presented be generalized to accommodate a wider range of LLMs and knowledge sources (not limited to factual triples, e.g. unstructured data, cultural context)?

4.There might be a need for further exploration into the long-term effects of repeated exposure to external knowledge distractors (EKDs) on LLMs.

---

> ### Author Rebuttal · Authors · 2024-05-29
>
> Thank you very much for your comments on our paper. Our work primarily takes an analytical approach to systematically explore the output patterns of models when external knowledge conflicts with parametric knowledge.
>
> > How can LLMs be better improved
>
> The conclusions we have drawn in Table 3 will promote LLM's alignment with knowledge and help design better methods for knowledge incorporation, enabling the model to trust external knowledge more (e.g., during knowledge updates) or to be less affected by external knowledge (when it is noise). For example, we can consider various aspects such as the length, format, and degree of knowledge changes and, based on practical application scenarios, encourage alignment between the model and internal or external knowledge.
>
> > Boundary of direct conflict
>
> We explore three different methods to introduce knowledge change. Among them, object distractor aligns the most with “direct conflict”, as our problem scenarios expect the model to answer the object. In our work, we distinguish “direct conflict” with others based on the sentence structure (subject / relation / object), as changing the answer (target object) of what is being asked is the most intuitive and direct way to introduce knowledge conflict. Building on our framework, future research can quantitatively and more granularly investigate the impact of knowledge conflict by changing the similarity between knowledge elements (e.g., altering meaningful noun entities or relations by different percentages).
>
> > Generalizability of our framework
>
> We believe that unstructured data ultimately consists of similar knowledge triplets. Therefore, the conclusions we have obtained can be partially transferred to these scenarios. Moreover, the knowledge types involved in our experiments are already diverse, providing inherent adaptability to different domains.

---

> > ### Comment · Reviewer_3wvY · 2024-06-05
> >
> > Thank you for your reply, I have no further questions. I think this paper is interesting and has a clear contribution. I will keep my scores.

---

### Decision · Program_Chairs · 2024-07-10

**Decision:**

Accept

**Comment:**

This paper studies the important problem of analyzing LLMs behavior under knowledge conflicts, an increasingly common setting in RAG and language agents. It proposes a parametric knowledge graph formulation, with which external knowledge extractors are then used in a controlled setting to examine different factors. Results with GPT-3.5 and MPT -7B show some interesting insights.

Reasons to Accept
- Analyzing LLMs' behavior under knowledge conflicts is an timely and important topic. This paper presents a study that seems more fine-grained, thanks to its parametric knowledge graph and distractors designs.
- Comprehensive experiments and analyses with two LLMs, with insights that may be informative to future studies.

Reasons to Reject
- The study is only based on two LLMs. It's not entirely clear which and to what extent will the observations generalize the different, especially stronger LLMs. Prior work (e.g., Xie et al., 2023) tends to use many more LLMs but focus on less (but more focused) aspects to analyze to improve the generalizability of the conclusions.
- The writing could be hard to follow at several places, as pointed out by a few reviewers.